# Carbon peaking prediction scenarios based on different neural network models: A case study of Guizhou Province

**Da Lian[1], Shi Qiang Yang[2], Wu Yang👤[3]\*, Min Zhang[3], Wen Rui Ran[4]**

1 China Railway Fifth Bureau Group Co., Ltd., Guiyang, China, 2 Geological Brigade of Guizhou Provincial Bureau of Geology and Mineral Resources, Zunyi, China, 3 Faculty of Resources and Environmental Engineering, Guizhou Institute of Technology, Guiyang, China, 4 Guizhou Natural Resources Survey and Planning Research Institute, Guiyang, China

\* yangwu@git.edu.cn

**Data Availability Statement:** All relevant data are within the paper and its Supporting Information files.

## Abstract

Global warming, caused by greenhouse gas emissions, is a major challenge for all human societies. To ensure that ambitious carbon neutrality and sustainable economic development goals are met, regional human activities and their impacts on carbon emissions must be studied. Guizhou Province is a typical karst area in China that predominantly uses fossil fuels. In this study, a backpropagation (BP) neural network and extreme learning machine (ELM) model, which is advantageous due to its nonlinear processing, were used to predict carbon emissions from 2020 to 2040 in Guizhou Province. The carbon emissions were calculated using conversion and inventory compilation methods with energy consumption data and the results showed an "S" growth trend. Twelve influencing factors were selected, however, five with larger correlations were screened out using a grey correlation analysis method. A prediction model for carbon emissions from Guizhou Province was established. The prediction performance of a whale optimization algorithm (WOA)-ELM model was found to be higher than the BP neural network and ELM models. Baseline, high-speed, and low-carbon scenarios were analyzed and the size and time of peak carbon emissions in Liaoning Province from 2020 to 2040 were predicted using the WOA-ELM model.

## Introduction

Global warming is a major issue for all countries, and one of the major strategies by which to relieve it is the reduction of greenhouse gases. In particular, the Fifth Assessment of Climate Change Bulletin of the United Nations Intergovernmental Panel on Climate Change (IPCC) objectively stated that climate warming is mainly caused by the burning of large amounts of fossil fuels during human activities, and that the alleviation of global warming has become an unavoidable responsibility of all countries. The increase in global average temperature caused by excessive $CO_2$ emissions seriously threatens the living space of human beings and sustainable development.

Global climate change is closely related to sustainable development goals worldwide, and consequently, many governments have taken relevant measures to actively address this issue.

**Funding:** This research was supported by Concealed Ore Deposit Exploration and Innovation Team of Guizhou Colleges and Universities (Guizhou Education and Cooperation Talent Team [2015]56), Provincial Key Discipline of Geological Resources and Geological Engineering of Guizhou Province (ZDXK[2018]001), Huang Danian Resources of National colleges and universities Teachers' Team of Exploration Engineering (Teacher Letter [2018] No. 1), Geological Resources and Geological Engineering Talent Base of Guizhou Province (RCJD2018-3), Key Laboratory of Karst Engineering Geology and Hidden Mineral Resources of Guizhou Province (Qianjiaohe KY [2018] No. 486Guizhou Institute of Technology Rural Revitalization Soft Science Project(2022xczx10), Education and Teaching Reform Research Project of Guizhou Institute of Technology (JGZD202107,2022TDFJG01).The funders had no role in study design, data collection and analysis, decision to publish, or preparation of the manuscript.

**Competing interests:** The authors have declared that no competing interests exist.

China has put forward the ambitious goal of "achieving peak carbon by 2030 and carbon neutrality by 2060." To help achieve this they have increased research and development into activities relating to new energy-technologies that can help reduce the proportion of fossil energy use, thereby achieving a profound change in the energy consumption structure. To protect the ecological environment, water, wind, and tidal energy resources should be utilized alongside "pollution and carbon reduction" activities. The guidance for typical demonstrations should be improved to fully mobilize the enthusiasm of local governments, departments, industries, and enterprises and thus facilitate good working patterns.

In China, Guizhou Province is a major karst area whose energy consumption is predominantly from coal, oil, and other primary energy sources. In recent years, accelerated urbanization and rapid economic growth have increased its dependence on traditional energy sources, and consequently, total energy consumption for this area is considered high. Furthermore, in Guizhou Province, the dual impetus of urbanization and economic development has lead to increased energy consumption and consequently, annually increasing carbon emissions. Energy saving and emission reduction strategies for Guizhou province will ultimately affect low-carbon economic development for China as a whole. To achieve peak carbon emissions in China by 2030, efficient emission reduction policies must be implemented. The impacts of carbon emissions must be studied and monitored using effective scientific methods to help enable accurate predictions. Different neural network models have been used to predict peak carbon for Guizhou Province as this will help to reduce carbon emissions to meet Chinas "3060" goal (Fig 1).

Existing research on carbon emissions has predominantly focused on national and industrial levels, while regional level research has been limited. This study focuses on Guizhou Province and the total carbon emissions were calculated using relevant data on energy consumption and carbon emissions. A literature review was conducted to identify the factors influencing carbon emissions, machine learning technology was used to identify high correlations among factors, and the future trends for carbon emissions were predicted. The future carbon emissions were then analyzed using the predominant factors and carbon peak data. The results of this study will help guide future theoretical research on carbon emissions at the regional level (Fig 1).

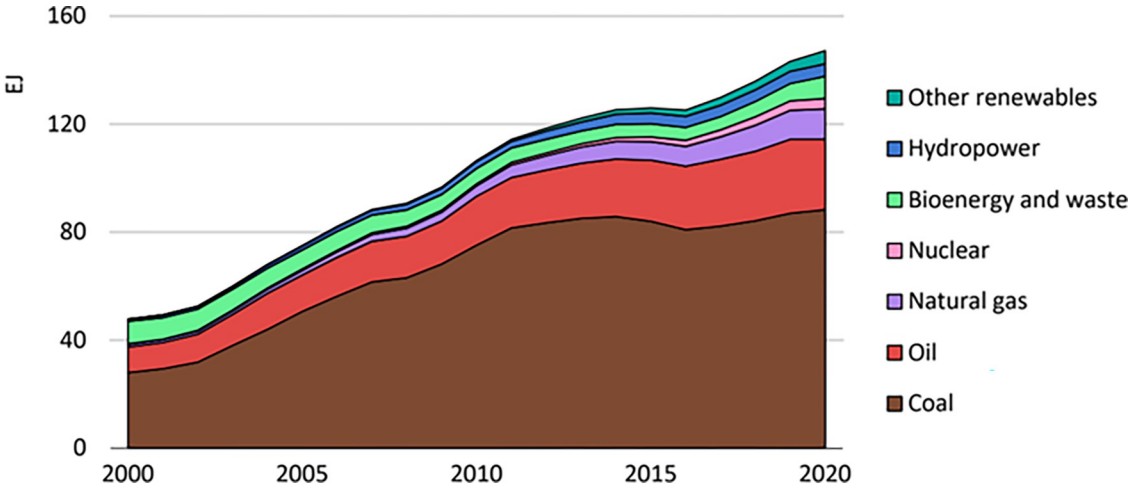

**Fig 1. Total primary energy demand by fuel in China.**

## Literature review

National carbon emission reduction work is required in all regions of a country, and regions with different levels of industrial development should implement differentiated policies. Guizhou Province is an old industrial base in China that predominantly used fossil fuel energy, and its greenhouse gas emissions, both historically and at a present, are generally high. Consequently, achieving Chinas carbon peak schedule will be difficult for Guizhou Province. A scientifically accurate calculation method for carbon emissions is thus required and will provide significant aid in the implementation of energy conservation and emission reduction strategies. Based on the optimized neural network model, this study predicts and analyzes the peak carbon emissions in Guizhou Province over the next 20 years, which helps to identify the existing problems and gaps and to guide the direction of energy development.

### Carbon emission influencing factors

Research on factors influencing carbon emissions is focused on the contributions of different factors and predominantly selects economic indicators such as population, economy, and energy intensity to construct a relevant index system. Ang et al. [1] analyzed the changes in carbon dioxide produced per unit of electricity globally and considered the import and export of each country, the fuel structure of power generation, and emission factors as the main influencing factors. This study has revealed that an improvement in power generation efficiency is the main reason for reduced $CO_2$ emissions. Rustemoglu et al. [2] studied carbon dioxide levels in Brazil and Russia from 1992 to 2012 and identified the factors affecting carbon emissions as falling into employment, economy, and carbon emission intensity categories. The results showed that Brazil's carbon dioxide emissions were not decoupled from its economic development. Russia's carbon dioxide emissions have greatly reduced with an increase in energy intensity. Lin et al. [3] used the input-output method to analyze the carbon emissions of the national food industry and divided industrial carbon emissions into four main factors: pollution factor, total output, energy intensity, and energy structure. Roinioti et al. [4] considered the development level of the national economy, energy consumption intensity, fuel consumption capacity, and carbon emission intensity as the main factors affecting carbon dioxide. Kim et al. [5] decomposed the factors affecting carbon emissions into production scale and production intensity and conducted corresponding research and analysis on the contributions of growth in energy consumption in various sub-industries. Roman et al. [6] focused on Colombia and used the IDA-LMDI model to decompose the influencing factors affecting $CO_2$ emissions into five aspects, energy intensity, wealth value, fossil fuel substitution, and renewable energy development, to explore the contribution levels of different $CO_2$ increments.

A carbon emissions impact factor index system was derived in China, predominantly from the perspectives of energy structure, population growth, economics, and other factors. Ying et al. [7] considered China's steel industry as the main research object, and their results showed that energy intensity has a greater impact on carbon emissions, but the role of the consumption structure was not as expected. Dewey et al. [8] studied the influencing factors of carbon dioxide generated by indirect consumption in the daily lives of Chinese residents and found that socioeconomic level is the main driving factor affecting $CO_2$ emissions for urban and rural residents, and the contributions of their different structures and their consumption proportions are inversely related to $CO_2$ emissions. Jingyan et al. [9] used regression analysis to study the carbon emissions of the thermal power industry in Guangdong Province, China. Whereas Ting et al. [10] used the LMDI method to conclude that economic growth and carbon dioxide levels change in the same direction; that is, the faster the

economic growth, the more obvious the effect of promoting carbon dioxide emissions, whereas the effective utilization and conversion of energy can reduce carbon dioxide emissions. Xiaoming et al. [11] used an LMDI decomposition model to study the carbon emissions from 30 provinces and cities in China from 2004 to 2014 and explore the contributions of the important factors by dividing the time period with 2009 as the demarcation point. The results showed that the growth of the gross national product had the greatest impact on national carbon emissions, while the contributions of other factors were weak. Ying et al. [12] used energy intensity, economic development, and population size as influencing factors to study carbon emissions. Xiaoyong et al. [13] used high-resolution spatial data to place carbonate chemical weathering carbon sinks, silicate chemical weathering carbon sinks, vegetation-soil ecosystem carbon sinks, and energy carbon emissions on a spatial grid. Subsequently, a carbon neutral index model was established to reveal the contributions of terrestrial ecosystem carbon sinks to carbon neutrality. The results were compared with those of other countries from horizontal and vertical perspectives. The results provide a new ideas for the measurement of carbon neutralization capacity, and provide important reference values and data for the systematic determination of global carbon neutralization capacity. the model developed by Xiaoyong et al. is highly recognized in academia.

## Domestic and foreign carbon emission predictions

Previous studies have predicted the carbon emissions for different countries or industries using logistic regression models, the STIRPAT model, and scenario analysis. Ouedraogo et al. [14] used the LEAP framework to model the analysis and projection of energy demand and associated emissions under alternative strategies in Africa from 2010 to 2040. Lin et al. [15] conducted a survey of China's manufacturing industry using the STIRPAT model and found that macroeconomic growth factors could determine the carbon dioxide emissions of the industry, while the effects of the fuel utilization and urbanization rates have significant regional heterogeneity. Wang et al.[16] constructed the STIRPAT model to investigate the factors influencing carbon emissions in Xinjiang from 1952 to 2012, and the results identified differences in the impacts of various factors in different historical periods. Prior to 1978, population size expansion was the main factor causing an increase in carbon emissions. From 1978 to 2000, economic growth and population size were the main factors driving increased carbon emissions, and after 2000, the main factors were increased economic development and fixed asset investment. Kachoee et al. [17] used the LEAP model to predict carbon dioxide emissions relating to Iran's power sector over the next 30 years and concluded that economic growth is the main influencing factor. Emodi et al. [18] used the LEAP model to study climate change in relation to Australia's power sector and found that reducing expenditure on environmental protection and resource conservation would produce economic benefits (Fig 2).Liu [19–22] designed four plane-scale models of steel oblique beam structures and conducted quasi-static tests under cyclic loading, which clarified the yield mechanism, failure mode, hysteresis energy consumption, stiffness degradation, equivalent viscous resistance coefficient and lateral deformation performance of oblique beam structures, and provided technical basis for performance-based seismic design of oblique beam structures [23]. Jun-song Jia [24] takes Henan Province of China as a study area, computed the EF and the ecological carrying capacity (EC) in 1949–2006. Based on the computed results, the simulating process of the ARIMA model and the fitting and forecasting results were explained in detail. The final results demonstrated that ARIMA model could be used effectively in the simulation and prediction of EF and the predicted EF could

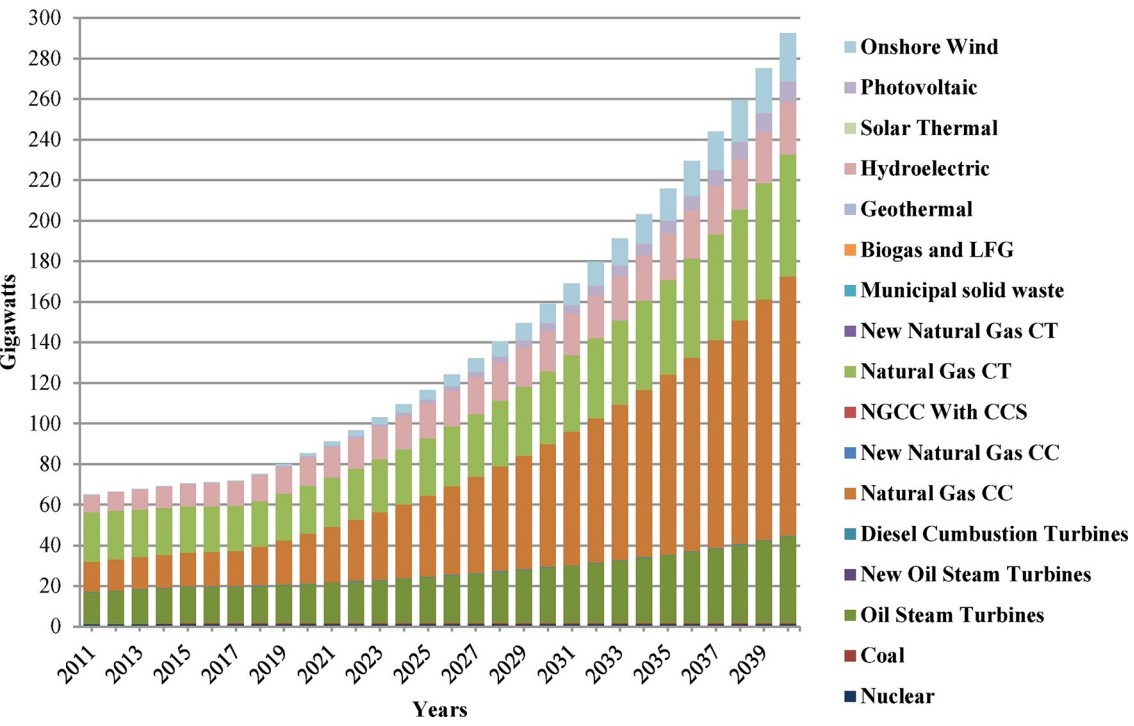

**Fig 2. Generation capacity with the BAU scenario until 2040.**

help the decision-makers make a package of better planning for regional ecological balance or sustainable future.

## Neural network models

Neural network models are widely used in various fields [25–27]. Representative neural network models include BP neural networks, radial basis function networks, Hopfield models, GMDH networks, adaptive resonance theory, Boltzmann machines, and CPN models. Lapedes et al. used neural networks for economic forecasting in 1987, whereas Chunjuan et al. [27] applied neural networks to predict typhoons, debris flows, and geological subsidence. Fan et al. [28] established a POS-BP neural network model to predict the total carbon emissions and intensities of 30 provinces, municipalities, and autonomous regions in China. Ying et al. [29] compared the advantages of a neural network and other traditional prediction methods and used a BP neural network model combined with a terminal information collection system and Web Service technology to design an intelligent system for urban road-occupying parking and proved the feasibility of the management system using actual data. Xiaowei et al. [30] predicted the prices of stock investments by combining a neural network model with principal component analysis and multiple linear regression. Xiaolong et al. [31] and others studied the problem of gas outbursts in tunnels using a BP neural network. The results were good and verified that the predicted and real values are consistent. Xiaocheng et al. [32] also used a BP neural network to predict air pollutant concentrations. The original BP neural network was used to calculate the system error of all samples using successive iterations and batch processing, which improved the execution efficiency.

In summary, the influencing factors affecting carbon emissions are predominantly considered to be population, economy, and energy structure, and these are used to establish a carbon emission index system and applied in the follow-up prediction research. Carbon emission

forecasting research has predominantly used traditional econometric methods such as the logistic regression model, STIRPAT model, and scenario analysis. Whereas neural network models have achieved satisfactory results when used for economic forecasting. A review of carbon emission-related literature shows that most recent studies have focused on the national or industrial levels, and few studies have focused on the use of machine learning algorithms for peak carbon emission predictions, while the reported neural network model used is relatively single. Training a single neural network prediction model is time-consuming and can easily fall into a local optimum. This study has thus combined a neural network model with carbon emission research and has optimized different algorithms to improve carbon emission predictions for Guizhou Province.

## Methods

Carbon emissions from Guizhou Province are calculated using an inventory method based with energy consumption data for the province. Referring to relevant literature and combining it with the actual development of Guizhou Province, 12 factors affecting carbon emissions were selected to establish a characteristic subset. By introducing the grey correlation analysis method, indicators with a higher degree of influence were selected and applied to follow-up prediction research. Finally, the carbon emissions from 2020 to 2040 in Guizhou Province were predicted under three different development scenarios by establishing a prediction model based on the WOA-ELM.

## Carbon emission calculations

There is no authoritative agency in China that directly provides carbon dioxide emission test data. This study has thus used a compromising method to discount the emission data for each year from different collected data and take the average value. The calculation of the carbon emissions and the collection of relevant data varied depending on the subjects of the study. Considering the characteristics of the carbon emissions in Guizhou Province and the difficulties regarding data acquisition, this study used an estimation method for carbon emissions proposed by the United Nations Intergovernmental Panel on Climate Change (IPCC) in 2006. This method is based on a gas emission inventory and calculates the carbon emissions generated by energy consumption by multiplying the energy consumption data by the energy carbon emission factor. The specific formula is as follows:

$$CO_2 = \sigma_1^n B_i \times C_i \times D_i \tag{1}$$

Where $B_i$ represents the consumption of the $i$th energy source and $n$ represents the type of energy source. In this study, n = 9 represents the energy consumptions of coal, coke, crude oil, gasoline, kerosene, diesel, natural gas, and electricity. The conversion and carbon emission coefficients for each energy standard coal in Guizhou Province are listed in Table 1.

From 2000 to 2012, the total carbon emissions from Guizhou Province increased (Table 2), while from 2012 to 2016 they showed a decreasing trend. The decrease is related to the construction of the ecological civilization in Guizhou Province, and the practice of green mountains is Jinshan and Yinshan. In 2020, the total amount of carbon emissions in Guizhou Province was 1.1 million tonnes 22237, approximately twice that in 2002. With the development of a social economy, carbon emissions in Guizhou Province are expected to show a steady upward trend in the future. However, due to the inhibition of carbon emissions through the implementation of policies, such as those for carbon emission reduction, the establishment of a carbon trading market, and an increase in the proportion of new energy applications in

**Table 1. Standard coal conversion and energy carbon emission coefficients.**

| Energy | Conversion coefficient for standard coal ($C_i$) | Carbon emission factor ($D_i$) |
|---|---|---|
| Coal | 0.7143 | 1.9003 |
| Coke | 0.9714 | 2.8604 |
| Crude oil | 1.4286 | 3.0202 |
| Gasoline | 1.4714 | 2.9251 |
| Kerosene | 1.4714 | 3.0179 |
| Diesel | 1.4571 | 3.0959 |
| Fuel oil | 1.4286 | 3.1705 |
| Natural gas | 1.33 | 2.1622 |
| Electricity | 0.1229 | 2.2132 |

Guizhou Province, the growth rate of carbon emissions will gradually decrease, and the development trend will be reduced year on year.

## Carbon emission influencing factors

Literature on carbon emission influencing factors shows that most scholars select macroeconomy, industrial structure, energy consumption, and scientific and technological development. Based on the actual social and economic development of Guizhou Province, this study has comprehensively considered the scientific, systematic, and authentic principles of index selection. Total population, urbanization rate, household consumption level, per capita GDP, energy intensity, carbon emission intensity, foreign direct investment, energy structure, proportion of primary secondary, and tertiary industries, and total energy consumption were selected and qualitatively analyzed. A description of each factor is as follows:

1. Total population. With the expansion of population size and the continuous improvement of living standards, the demand for food, clothing, shelter, and living environments increases. This drives the development of related industries, and an increase in energy demand, which leads to increased $CO_2$ emissions.

2. Urbanization rate: The urbanization rate reflects the progress of urbanization in a country or region. Improvement in the urbanization rate can drive an increase in economic development, thus promoting the process of local industrialization. Further promotion of

**Table 2. Guizhou Province carbon emissions from 2000 to 2020 (converted value).**

| Year | Total carbon emissions/10,000 tonnes | Year | Total carbon emissions/10,000 tonnes |
|---|---|---|---|
| 2000 | 11800.5 | 2012 | 21062.6 |
| 2001 | 12126.2 | 2013 | 23644.6 |
| 2002 | 14365.3 | 2014 | 25676 |
| 2003 | 14713.3 | 2015 | 25458.8 |
| 2004 | 14857.5 | 2016 | 22446.9 |
| 2005 | 19037.3 | 2017 | 24327.4 |
| 2006 | 18245.7 | 2018 | 22986.3 |
| 2007 | 18411.2 | 2019 | 22080.7 |
| 2008 | 16954 | 2020 | 22237.1 |
| 2009 | 15587.9 | | |
| 2010 | 16789.4 | | |
| 2011 | 21733.7 | | |

industrialization in Guizhou Province will inevitably generate a large energy demand. If we only pay attention to the development of urbanization and ignore resources and the environment, it will inevitably lead to large levels of energy consumption and the vicious growth of carbon dioxide.

3. Resident consumption levels: Increasing resident consumption levels accelerates the pace of urban industrialization and rapid economic growth.

4. GDP per capita: Per capita GDP generally reflects a country's economic development and people's living standards, but at the same time, it can also express the growth level of carbon emissions caused by the continuous expansion of economic development and scale to a certain extent. Economic and social development cannot be separated from industrial development, and industrial development cannot be separated from energy. The rapid development of the global economy will inevitably lead to increased energy consumption, resulting in high levels of carbon dioxide emissions.

5. Energy intensity: The amount of energy consumed per unit of output reflects the dependence of social and economic development on energy consumption. A lower energy intensity implies lower energy use, higher energy efficiency, and increased output [33].

6. Carbon emission intensity: Carbon emission intensity refers to the proportional relationship between total carbon emissions and gross domestic product (GDP), which shows the relationship between the social and economic development of a country or region and carbon emissions. Generally, the better the economic and social development, the higher the level of industrial development and the lower the carbon emission intensity [34]. With economic development and progress in science and technology, the intensity of carbon emissions will gradually decrease.

7. Foreign direct investment: Since the reform and opening-up, the speed of introducing and utilizing foreign direct investment in Guizhou Province has increased. However, environmental problems have become increasingly severe with economic development. Considering the negative impact of foreign direct investment on carbon emissions and other environmental problems in the region, it is necessary to study the impact of foreign direct investment on carbon emissions in Guizhou Province.

8. Energy structure: The proportion of coal consumption has a significant impact on the total carbon emissions [35]. The ratio of coal energy consumption to fossil fuel energy consumption was set to represent the energy structure.

9. Primary industry: The primary industry is the production sector, with agriculture as its main object and nature as its direct object. The higher the proportion of the primary industry, the lower the degree of industrial modernization in the region. During continuous economic development, the proportion of primary industry gradually decreases in relation to the larger industrial structure.

10. Secondary industry: Secondary industry includes industrial enterprises and includes numerous sub-industries have high energy consumption and pollution levels. Many links promote the increase of energy consumption, resulting in a large amount of carbon dioxide emissions [36]. Selecting the proportion of secondary industry can better show the level of the industrial structure in Guizhou Province.

11. Tertiary industry: As an industrialized city, the proportion of tertiary industry in Guizhou Province has gradually increased in recent years. The rational transformation of the

industrial structure will promote low-carbon technological innovation and have a significant positive impact on reducing carbon emissions.

12. Total energy consumption: Energy is important for economic development and energy consumption plays a decisive role in economic growth. Different energy structures cause different degrees of environmental pollution [37].

The selected indicators are shown in Table 3.

## Results

Total carbon emissions from Guizhou Province between 2000 and 2020 were calculated as basic data, and correlations with the population, economy, and energy data were determined. The data were obtained from the Statistical Yearbook of Guizhou Province 2000–2020.

To facilitate subsequent modeling and data representation, the variable names for the 12 factors affecting carbon emissions were redefined as X1, X2, X3, X4, X5, X6, X7, X8, X9, X10, X11, and X12. The total carbon emissions (Y) from Guizhou Province were set as the reference sequence, and the 12 influencing factors are set as the comparison sequence. The original data were normalized to eliminate dimensional influences, and the results are shown in Table 4.

To calculate the difference between the maximum and minimum absolute values in the matrix, the resolution coefficient was set to 0.5 to obtain the correlation coefficient table. The average values of the correlation coefficients for different sequences at each time point were used to obtain the correlation degree and for sorting. The results are summarized in Table 5.

The closer the correlation degree is to 1, the stronger the correlation with carbon emissions in Guizhou Province. The top five correlation degrees identified were for X12, X1, X2, X4, and X3. The influencing factors were total energy consumption, total population, urbanization rate, per capita gross domestic product (GDP), and residents' consumption level. The correlation degrees, which were the main correlation factors, were all greater than 0.75. The correlation degrees for X7, X11, X10, X9, X8, and X5 were all between 0.5 and 0.7, indicating medium correlation. The correlation coefficient between energy intensity and carbon emissions in Guizhou Province was low at 0.487.

The relevant data regarding energy consumption in Guizhou Province were collected and the carbon emission data between 2000 and 2020 were determined. The results show that the total carbon emissions in Guizhou Province are closely related to economic development and relevant policies. Data from the existing literature was used to help analyze the data from

Table 3. Influencing factors and indicators of carbon emission in Guizhou Province.

| Indicator layer | Unit |
|---|---|
| Total population | Ten thousand people |
| Urbanization rate | % |
| Residents' consumption level | CHY/person |
| GDP per capita | CHY/person |
| Energy intensity | Ton of standard coal/10,000 CHY |
| Carbon emission intensity | Ton/10,000 CHY |
| Foreign direct investment | Ten thousand dollars |
| Energy structure | % |
| Proportion of primary industry | % |
| The proportion of the secondary industry | % |
| Proportion of tertiary industry | % |
| Total energy consumption | Ten thousand tonnes of standard coal |

**Table 4. Normalization results for the indicator data.**

| Year | Y | X1 | X2 | X3 | X4 | X5 | X6 | X7 | X8 | X9 | X10 | X11 | X12 |
|------|------|------|------|------|------|------|------|------|------|------|------|------|------|
| **2000** | 0 | 0 | 0 | 0 | 0 | 1 | 0 | 1 | 0.65 | 0.04 | 0.65 | 0.09 | 0.04 |
| **2001** | 0.04 | 0.08 | 0.01 | 0.02 | 0.01 | 0.86 | 0.01 | 0.94 | 0.59 | 0.04 | 0.59 | 0.1 | 0.04 |
| **2002** | 0.08 | 0.16 | 0.03 | 0.03 | 0.03 | 0.7 | 0.03 | 0.93 | 0.59 | 0.03 | 0.59 | 0.2 | 0.03 |
| **2003** | 0.06 | 0.23 | 0.04 | 0.04 | 0.04 | 0.59 | 0.03 | 0.92 | 0.54 | 0 | 0.54 | 0.22 | 0 |
| **2004** | 0.09 | 0.28 | 0.05 | 0.06 | 0.05 | 0.55 | 0.02 | 0.66 | 0.55 | 0.02 | 0.55 | 0.27 | 0.02 |
| **2005** | 0.22 | 0.32 | 0.06 | 0.07 | 0.07 | 0.56 | 0.04 | 0.64 | 0.68 | 0.11 | 0.68 | 0.24 | 0.11 |
| **2006** | 0.17 | 0.36 | 0.07 | 0.09 | 0.09 | 0.5 | 0.06 | 0.4 | 0.58 | 0.11 | 0.58 | 0.34 | 0.11 |
| **2007** | 0.23 | 0.39 | 0.09 | 0.1 | 0.1 | 0.44 | 0.09 | 0.56 | 0.54 | 0.1 | 0.54 | 0.37 | 0.1 |
| **2008** | 0.31 | 0.41 | 0.1 | 0.1 | 0.13 | 0.41 | 0.15 | 0.8 | 0.57 | 0.15 | 0.57 | 0.37 | 0.15 |
| **2009** | 0.42 | 0.44 | 0.13 | 0.12 | 0.15 | 0.44 | 0.14 | 0.72 | 0.43 | 0.28 | 0.43 | 0.41 | 0.28 |
| **2010** | 0.55 | 0.45 | 0.55 | 0.16 | 0.21 | 0.34 | 0.08 | 0.82 | 0.56 | 0.31 | 0.56 | 0.35 | 0.31 |
| **2011** | 0.61 | 0.62 | 0.56 | 0.18 | 0.26 | 0.31 | 0.17 | 0.77 | 0.61 | 0.41 | 0.61 | 0.34 | 0.41 |
| **2012** | 0.59 | 0.72 | 0.57 | 0.23 | 0.33 | 0.27 | 0.28 | 0.59 | 0.65 | 0.52 | 0.65 | 0.31 | 0.52 |
| **2013** | 0.67 | 0.78 | 0.6 | 0.31 | 0.43 | 0.2 | 0.38 | 0.45 | 1 | 0.6 | 1 | 0 | 0.6 |
| **2014** | 0.73 | 0.87 | 0.61 | 0.36 | 0.48 | 0.19 | 0.51 | 0.33 | 0.78 | 0.69 | 0.78 | 0.23 | 0.69 |
| **2015** | 0.85 | 0.99 | 0.68 | 0.46 | 0.61 | 0.14 | 0.7 | 0.17 | 0.9 | 0.82 | 0.9 | 0.14 | 0.82 |
| **2016** | 0.94 | 1 | 0.76 | 0.58 | 0.75 | 0.1 | 0.83 | 0.11 | 0.94 | 0.94 | 0.94 | 0.12 | 0.94 |
| **2017** | 1 | 0.99 | 0.82 | 0.69 | 0.85 | 0.07 | 0.92 | 0 | 0.85 | 1 | 0.85 | 0.2 | 1 |
| **2018** | 0.96 | 0.95 | 0.85 | 0.78 | 0.94 | 0.01 | 1 | 0.31 | 0.82 | 0.87 | 0.82 | 0.23 | 0.87 |
| **2019** | 0.93 | 0.93 | 0.88 | 0.88 | 1 | 0 | 0.94 | 0.26 | 0.68 | 0.88 | 0.68 | 0.39 | 0.88 |
| **2020** | 0.87 | 0.86 | 0.91 | 0.95 | 1 | 0 | 0.14 | 0.13 | 0.4 | 0.87 | 0.4 | 0.63 | 0.87 |

Guizhou Province, this study initially sets 12 indicators, including the total population, urbanization rate, consumption level of residents, and per capita GDP, as the factors influencing carbon emissions and expounds the reasons for the selection in detail. The total energy consumption, total population, urbanization rate, per capita GDP, and residents' consumption level all showed a high level of correlation with carbon emissions in Guizhou Province, and the five factors with strong correlation can be used as input variables in the prediction model to improve carbon emission prediction accuracy in Guizhou Province.

**Table 5. Ranking the correlation degrees of the influencing factors.**

| Influencing factors | Correlation degree X | Incidence order |
|---------------------|----------------------|-----------------|
| **X1** | 0.8573 | 2 |
| **X2** | 0.8534 | 3 |
| **X3** | 0.7589 | 5 |
| **X4** | 0.7953 | 4 |
| **X5** | 0.5046 | 12 |
| **X6** | 0.4870 | 11 |
| **X7** | 0.6907 | 6 |
| **X8** | 0.5060 | 10 |
| **X9** | 0.5166 | 9 |
| **X10** | 0.6632 | 8 |
| **X11** | 0.6968 | 7 |
| **X12** | 0.8801 | 1 |

## Carbon emission and carbon peak prediction model

### Prediction model design

**Establishment of a BP neural network model.** A BP neural network is composed of input, hidden, and output layers. When establishing a BP neural network, setting too many or too few hidden layer nodes will affects the results of the data; too many hidden layers are prone to overfitting, resulting in increased training time, and too few hidden layer nodes affects the accuracy of the data fitting. The number of hidden layers must be determined based on data characteristics. In this study, the number of hidden layer nodes [38] was determined by trial and error. The specific settings for each level and node are as follows:

1. *Setting the model input and output layers*. The five factors selected in the third chapter are the main factors affecting carbon emissions in Guizhou Province, and were used as the model input variables; that is, the nodes of the input layer, including the total population, urbanization rate, household consumption level, per capita GDP, and total energy consumption. The number of neurons in the output layer of the model was one, that is, the carbon emissions of Guizhou Province.

2. *Analysis of network structural parameters*. ***a) Selection of network layer number.*** A BP neural network is a multilayered neural network, and the selection of the number of layers is crucial for establishing a reasonable network model. Different numbers of hidden layers affected the final model-fitting. When the number of hidden layers is increased, the network structure becomes complex, the prediction ability of the model is improved, and the error is reduced. However, the overfitting phenomenon is prone to occur, and the training time is not ideal. The number of hidden layers is reduced, the network structure is simplified, the prediction ability of the model is reduced, the accuracy is reduced, the error is increased, and the training speed is fast. Hornik proved that a three-layer BP network structure can meet the fitting requirements of most nonlinear systems; if the number of hidden layer nodes is set appropriately, it can also help the network fit most functions with a high level of accuracy. To meet the requirements for training accuracy, a three-layer BP neural network with only one hidden layer was selected for this study to predict carbon emission data.

### b) Determination of the activation function and other parameters

The activation function introduced a nonlinear relationship into the neurons through mapping. To better represent the nonlinear relationship of a function, an appropriate type of activation function must be selected. The hyperbolic tangent function is a common activation function, which maps the number taking the value of $(-\infty, +\infty)$ into $(-1, 1)$, so that the variable is in the largest possible threshold range, which can better preserve the nonlinear variation level of the function. The transfer function of the hidden layer node was thus chosen as the tangent S-type transfer function, tansig, for this study. The input and output values of the linear transfer function purelin can assume any value. To facilitate comparisons with the sample value, purelin was selected as the output value returned by the output layer, which refers to the change in the information accumulation speed of the BP neural network with time. Different learning rate settings affect the training time and the training effect of the model. The training speed of the model with a larger learning rate value is relatively fast; however, there are large fluctuations in the later period that the resulting model cannot convert. When its value is small, although it can make the simulation results of the model more accurate, it significantly increases the training time. In general studies, the learning rate $\gamma$ is usually set between 0 and

1. In this paper, through continuous debugging and comparison of the training effect in the training process, the learning rate $\gamma = 0.1$ was selected. The accuracy of the network training was required to be 0.001, and the maximum number of training sessions was 500.

### c) Selection of the number of neurons in the hidden layer

The number of neurons in the hidden layer affects the network performance. The more hidden layer nodes, the more complex the model training, the more time-consuming the process, and it may not be able to achieve function approximation smoothly, resulting in a shock effect. If the number of neurons in the hidden layer is too small, the training accuracy of the network may not meet the ideal requirements, resulting in experimental result failure. After defining the relevant network parameters and activation functions, the approximate range was calculated using an empirical formula, and the number of hidden layer neurons was determined by debugging a single variable trial and error. According to the existing empirical formula, the following three quantitative relationships exist.

$$k = 2m + 1 \tag{2}$$

$$k = log2m \tag{3}$$

$$k = m + n + a \tag{4}$$

Where K is the number of nodes in the hidden layer, m is the number of nodes in the input layer, n is the number of nodes in the output layer, and a is an arbitrary constant between 1 and 10. This study used an empirical formula to calculate the number of nodes in the hidden layer, which can be selected between 4 and 13. After adjusting the parameter settings several times and combining them with the overall level of the training results, as listed in Table 6, the number of nodes in the hidden layer was set to 10. Finally, the number of neurons in the input layer of the BP neural network used in this study was 5, the number of nodes in the hidden layer was 10, the number of nodes in the input layer was 1, the learning rate was 0.1, and the setting accuracy was 0.001.

### Establishing an extreme learning machine model

The algorithm can be divided into three steps. The first step determines the number of neurons in a hidden layer and randomly generates a connection value between an input layer and the hidden layer, and a neuron bias for the hidden layer in a network model. The second step

**Table 6. Effects of different hidden layer node numbers on the model.**

| Implies the number of layer nodes | Number of training sessions | Mean absolute percent error |
|:---:|:---:|:---:|
| 4 | 200 | 7.25 |
| 5 | 102 | 8.04 |
| 6 | 133 | 6.36 |
| 7 | 56 | 10.04 |
| 8 | 74 | 9.28 |
| 9 | 120 | 6.03 |
| 10 | 42 | 4.37 |
| 11 | 86 | 5.41 |
| 12 | 67 | 6.79 |
| 13 | 50 | 11.26 |

determines the activation function of the neurons in the hidden layer, and calculates the output matrix H for the hidden layer by selecting an infinitely differentiable function. The third step calculates the output layer weight as follows: $\beta^* = H + T$.

## Establishment of a WOA-BP model

The BP algorithm is a common learning algorithm used in various fields. However, existing problems restrict its development. During the training process, the initial weights and thresholds are randomly generated, and consequently, the generalization ability cannot be guaranteed. The WOA is then used to optimize the initial parameters of the BP neural network to obtain a more stable WOA-BP neural network.

The steps involved in the WOA optimization of the BP neural network are as follows:

1. Determine a BP neural network structure and initial weight and threshold values;

2. Calculate that individual fitness of the whale, and take a fitness function as an optimized target function;

3. Set an algorithm-stopping criterion, select different mechanisms, update the individual positions of the whale, and optimize parameters;

4. Identify an optimal whale position, and assign the optimal weight and threshold values to the BP neural network;

5. The optimized BP neural network is trained, a simulation test is performed, and the prediction performance of the BP neural network is compared with the data prior to optimization.

## Establishment of a WOA-ELM model

Based on the whale optimization algorithm and structure of the extreme learning machine mentioned above, a WOA-ELM combination forecasting model was established. In the WOA, the optimal position of the humpback whale is the optimized ELM parameter value, and the WOA iteration is used to determine the optimal wi and bi of the ELM, which can improve the prediction accuracy of the model.

1. Initialize the ELM: Number of input neurons (set as 5), number of hidden layer neurons (set as 10), activation function type G (set as a sigmoid function), input weight (wi), and hidden layer threshold (bi);

2. Initialize the WOA parameters: Population size N (set to 50) and maximum generation number Tmax (set to 500);

3. Initialize the position vector of the individual whale, connecting the randomly generated connection weight (wi) and nerve during ELM training, the meta-bias (bi) was considered the initial position vector of the individual whale;

4. Set a fitness function (error rate in the ELM test) and calculate the current individual fitness in the initialized population to obtain the optimal (best fitness) whale individual.

5. After the p-value is randomly generated (0, 1), the updated formula (Graph) (Graph) at different positions is determined by the values of A and p. When the A (Graph) (Graph) values are different, there are three corresponding update position formula selections as follows: When A (Graph) (Graph) > 1, select to perform a global random search; when A (Graph)

(Graph) < 1, the random variable p-value is combined to choose between the shrinking enclosure and spiraling strategies.

6. Determine whether the algorithm can satisfy the preset termination conditions. When the end condition of the algorithm is reached, the algorithm is terminated, and the whale individual position vector with the best fitness value is output; that is, the optimal weight and threshold for the ELM network are obtained. Otherwise, the number of iterations is increased by one, and Step (5) is repeated.

7. The obtained optimal parameters of the ELM network are input into the WOA-ELM model to predict the carbon emissions for Guizhou Province.

It is worth noting that in ELM, the input data is transformed by the hidden layer, and then the output layer produces the result. This process is "forward propagation", that is, information flows from the input layer to the output layer. However, the most important thing in this process is backpropagation, that is, how to adjust network parameters to improve performance when the output does not meet expectations. Back-propagation algorithm is an important optimization technique, which calculates how much the weight of each layer needs to be adjusted according to the difference between the actual output and the expected output of the network, and then optimizes the network. In ELM, due to its single-layer feedforward feature, backpropagation is mainly used to adjust the weights and biases, so that the network can better adapt to the training data.

## Analysis and comparison of the experimental results from the prediction model

There is no authoritative agency in China that directly provides carbon dioxide emission data, and consequently, this study has used a compromise method to discount the emission data for each year from different database collections and obtain an average value. The calculation of carbon emissions and the collection of relevant data varied depending on the subjects of the study. Considering the characteristics of carbon emissions in Guizhou Province and the difficulty in acquiring data, this study has used an estimation method.

## Prediction based on BP neural network model

1. *Simulation settings*. The data used in this study were obtained from the Statistical Yearbook of Guizhou Province from 2000 to 2020, the China Energy Statistics Yearbook, and the website of the National Bureau of Statistics. In the third chapter, the calculation of carbon emissions data in Guizhou Province, for example, to verify the BP neural network prediction model data for carbon emissions in Guizhou Province, the fitting degree of the evaluation model and its advantages and disadvantages are discussed.

2. *Prediction results and analysis*. When dividing the training and test sets, relevant data from 2015 to 2020 were used as the training set data, and the remaining six groups were used as the test set data. Carbon emissions change each year, and thus, to improve the prediction performance of the model, this study has predicted carbon emissions annually and added the influencing factors and carbon emission data from each new prediction year to the training sample. The relative and absolute errors were used to analyze the prediction effect of the prediction model. The prediction errors for the test samples are shown in Table 7.

**Table 7. BP neural network prediction model errors.**

| Year | Actual value/ 10 000 t | Predicted value/ 10 000 t | Relative error (%) | Absolute error/ 10 000 t |
|------|----------------------|--------------------------|-------------------|-------------------------|
| 2015 | 25458.8 | 22667.6 | 10.96 | 2791.2 |
| 2016 | 22446.9 | 22913.9 | −2.08 | −467 |
| 2017 | 24327.4 | 24815.2 | −2.01 | −487.8 |
| 2018 | 22986.3 | 22932.2 | 0.24 | 54.1 |
| 2019 | 22080.7 | 22457.6 | −1.71 | −376.9 |
| 2020 | 22237.1 | 21234.3 | 4.51 | 1002.8 |

The carbon emission prediction curve in Table 7 shows that the change rule for the predicted carbon emission values in Guizhou Province was generally consistent with the change rule for the real values; however, the difference between the values is large, and the prediction effect is not sufficiently stable. The difference between the predicted and real values in 2018, 2019, and 2020 was small, and the relative error was below 2.5%, which met the expectations for prediction accuracy. However, there were large differences between the predicted and actual values in 2015, 2016, and 2017, and the expected prediction effects were not observed. This was predominantly because the initial weights and thresholds in the BP neural network are determined randomly, and it is difficult to achieve a good fit during model training; this results in large fluctuations in the prediction results and an inability to achieve a good prediction effect [39–42].

## Extreme learning machine model predictions

An extreme learning machine model was used to predict carbon emissions for Guizhou Province, using the data from 2000 to 2014 as the training set. The data from 2015 to 2020 were divided into the test set, and the training sample method was the same as for the BP neural network structure, the relative error was reserved for two digits, the absolute error was reserved for one digit, and the error comparison between the actual value and the predicted value was obtained.

The prediction results in Table 8 show that the model fits the carbon emissions of Guizhou Province well and also approximates their relationship with the influencing factors. However, the predicted results are unstable. Although the predicted values for most years were close to the actual values, the carbon emissions obtained in 2017 differed significantly from the actual values. In the forecast results, the absolute error between the forecast and real values in 2019 was 8.8, and the forecast value in 2019 was the closest to the actual carbon emissions of Guizhou Province. The average relative error of the test set was 0.43%. Compared with the BP neural network, the prediction model for carbon emissions in Guizhou Province based on

**Table 8. Extreme learning machine model prediction errors.**

| Year | Actual value/ 10 000 t | Predicted value/ 10 000 t | Relative error (%) | Absolute error/ 10 000 t |
|------|----------------------|--------------------------|-------------------|-------------------------|
| 2015 | 25458.8 | 25405.3 | 0.21 | 53.46348 |
| 2016 | 22446.9 | 22229.1 | 0.97 | −217.73493 |
| 2017 | 24327.4 | 24030.6 | 1.22 | −296.79428 |
| 2018 | 22986.3 | 22974.8 | 0.05 | 11.49315 |
| 2019 | 22080.7 | 22071.8 | 0.04 | -8.83228 |
| 2020 | 22237.1 | 22203.7 | 0.15 | 33.35565 |

extreme machine learning has a higher accuracy and stronger ability to approximate the non-linear relationship of samples, but the setting of random initialization value and β also affects the accuracy of the model to a certain extent, and will require further optimization.

## WOA-BP model predictions

The whale algorithm, which has a global search ability, was used to optimize the initial weight threshold of the BP neural network to improve its prediction accuracy. The divided training and test sets were the same as those used in the BP neural network model.

When setting the initial weights and thresholds of the neural network, a set of randomly generated initial values was selected, because there was no relevant setting principle. The BP neural network can learn the mapping relationship between the input and output automatically, generate initial parameters randomly, and modify the weights and thresholds of the network continuously through error back propagation; however, randomly selected initial weights and thresholds are usually inversely proportional to the convergence speed of neural network training; that is, the larger the value, the slower the convergence speed. In this case, the final training results easily fall into the local optimum, and it is difficult to obtain ideal calculation and prediction results. As shown in Table 9, the relative error of the BP neural network after optimization is significantly reduced by no more than 1.5%, and the fitting degree for carbon emissions in Guizhou Province is significantly higher when compared with the results prior to optimization. The carbon emission prediction value in 2017 was the closest to the actual value, with a relative error of 0.16%, and the prediction results in other years were relatively stable. The accuracy and stability of the prediction using the WOA-BP neural network were significantly improved [43, 44].

## WOA-ELM model predictions

The training samples selected in this section were the same as those used for the extreme learning machine model setting. The input and output variable data from 2000 to 2014 were used as the training set (Table 10), and the prediction years were from 2015 to 2020. The WOA-ELM model was established [45–50].

The results show that after multiple training sessions, the ELM model has a better fitting effect on the carbon emissions of Guizhou Province from 2015 to 2020 after WOA optimization. The error between the predicted value and the real value is between 0% and 0.05%. The relative error and absolute error between the two are relatively small, except for a few, and the prediction accuracy is significantly higher than that of the ELM model. The effectiveness of the WOA-optimized ELM scheme was verified.

## Comparison of the prediction results

In this study, four prediction models were established to test the carbon emission data for Guizhou Province. To evaluate the prediction performance of the four models, three indicators,–

**Table 9. WOA-BP prediction model errors.**

| Year | Actual value/ 10 000 t | Predicted value/ 10 000 t | Relative error (%) | Absolute error/ 10 000 t |
|------|------------------------|---------------------------|--------------------|--------------------------|
| 2015 | 25458.8 | 25260.2 | 0.78 | 198.57864 |
| 2016 | 22446.9 | 22170.8 | 1.23 | −276.09687 |
| 2017 | 24327.4 | 24288.4 | 0.16 | −38.92384 |
| 2018 | 22986.3 | 22834.5 | 0.66 | 151.70958 |
| 2019 | 22080.7 | 21928.3 | 0.69 | −152.35683 |
| 2020 | 22237.1 | 22119.2 | 0.53 | 117.85663 |

**Table 10. WOA-ELM prediction model errors.**

| Year | Actual value/ 10 000 t | Predicted value/ 10 000 t | Relative error (%) | Absolute error/ 10 000 t |
|------|------------------------|---------------------------|--------------------|--------------------------|
| 2015 | 25458.8 | 25458.8 | 0 | 0 |
| 2016 | 22446.9 | 22303.23984 | 0.64 | 143.66016 |
| 2017 | 24327.4 | 24215.49396 | 0.46 | 111.90604 |
| 2018 | 22986.3 | 22979.40411 | 0.03 | 6.89589 |
| 2019 | 22080.7 | 22078.49193 | 0.01 | 2.20807 |
| 2020 | 22237.1 | 22225.98145 | 0.05 | 11.11855 |

mean absolute error, mean absolute percentage error, and root mean square error were used for comparative analysis. The mean absolute error indicates the actual prediction error, while the root mean square error (RMSE) indicates the deviation between the observed and true values.

The results showed that the BP neural network is highly accurate at predicting carbon emission data for Guizhou Province using an extreme learning machine model (Table 11). Compared with the ELM model, the BP neural network model is less robust and random, and its convergence speed is slightly lower than that of the ELM model. Among the four prediction models, the prediction model based on the WOA-ELM had the highest prediction accuracy, and it was followed by the model based on the extreme learning machine. The prediction performance of the BP neural network model was the worst. The comparison test results concluded that the prediction effect of the WOA-ELM model was relatively better, and that the prediction accuracy can reach the expected level, which can be used to predict the peak carbon emissions of Guizhou Province in the following text.

## Carbon peak predictions for Guizhou Province

### Construction of carbon emission scenarios

**Scenario settings.** Scenario analysis refers to the quantitative analysis of both past and present situations, and integrates the factors affecting the future and makes qualitative assumptions to infer possible future situations. It is not the purpose of scenario construction to accurately predict the possibilities of the future, as its greatest practical value is comprehensive analysis. When using scenario analysis, there are two premises: one is to ensure that impact factors can be quantified and the other is to predict future indicators [33–35, 51, 52].

This section uses a scenario analysis method to set the impact factors of carbon emissions under different development scenarios as this will help to facilitate theprediction of carbon emission levels in Guizhou Province from 2020 to 2040. First, the baseline, high-speed, and low-carbon scenarios are set, corresponding to the indicators of medium growth and high growth with positive regression coefficients. Then, according to strategic policy interpretation for economic and energy development in Guizhou Province, the current situation for

**Table 11. Prediction accuracies of the different models.**

| Model category | BP neural network model | ELM model | WOA-BP model | WOA-ELM model |
|----------------|-------------------------|-----------|--------------|---------------|
| MAE | 575.93 | 234.46 | 349.65 | 101.13 |
| MAPE | 1.08% | 0.44% | 0.63% | 0.21% |
| RMSE | 761.6 | 338.65 | 387.16 | 166.16 |

(MAE:Mean Absolute Error;MAPE:Mean Absolute Percentage Error;RMSE:Root Mean Square Error)

**Table 12. Profile setting description.**

| Profile | Total population | Urbanization rate | Residents' consumption level | GDP per capita | Total energy consumption |
|---|---|---|---|---|---|
| Baseline scenario | Medium | Medium | Medium | Medium | Medium |
| High-speed scene | High | High | High | High | High |
| Low-carbon scenarios | Medium | Medium | High | Medium | Medium |

economic and social development and the development trend for the energy structure in Guizhou Province were clarified, and the parameters for total population, urbanization rate, residents' consumption level, total energy consumption, and per capita GDP in Guizhou Province in the future under different development scenarios were set in combination with relevant policies and energy target requirements. Finally, the future evolutionary trend for carbon emissions in Guizhou Province was predicted (Table 12).

*Benchmark scenario*. The benchmark scenario is the continuation of existing economic and energy development in Guizhou Province. In the current economic development model, each factor is set according to the most likely situation. As a large industrial province, Guizhou Province has a complete industrial infrastructure that is expected to continue to thrive. Furthermore, its economic and industrial structures will continue to follow the state calls for transformation and upgrading. The energy consumption structure of Guizhou Province is dominated by industrial development, and will continue to be dominated by fossil energy consumption; however, the proportion of fossil energy consumption will continue to decline with the development of new energy technologies.

*High-speed scenario mode*. In the high-speed scenario mode, the total population, urbanization rate, per capita GDP, household consumption level, and total energy consumption are maintained, which facilitates rapid development and change. With the rapid growth of the population, acceleration of urbanization, rapid and vigorous development of the economy and society, rapid development of new industries, and dominant position of the information industry, the use of new energy will be more widely applied in various industries, and the efficiency of energy utilization will be significantly improved.

*Low-carbon scenario*. Total population, urbanization rate, per capita GDP, and total energy consumption will develop at a lower rate than in the baseline scenario.

## Scenario parameter settings

*1) Population setting*. With economic and social development, the total population will continue to expand in the short term, however, long term, the population growth rate will decline. Analysis of the changes in the total population trend in Guizhou Province showed that it gradually decreased from 2010 to 2020, and the natural growth rate of the population was negative. By 2020, the population of Guizhou Province will reach 38.57 million, while the Population Development Plan of Guizhou Province proposes that the permanent population will reach 50 million by 2030; which means that the average annual growth rate will be 0.44%. According to the population development plan of Guizhou Province and the population growth in recent years, this study sets the annual rate of change at 0. 7% in baseline mode, 1% in high-speed mode, and 0. 5% in the low-carbon mode.

*2) Setting the urbanization rate*. Urbanization is continuously advancing in Guizhou Province, reaching a rate of 50.26% in 2018, which was 2.58 times higher than that in 1995. The average growth rate in the past five years was 1.12%, and the average growth rate in the past ten years was 1.33%. The effects of changing urbanization trends on the economies of various countries show that Britain and the United States are in the leading position in the process of

global urbanization construction, reaching approximately 80%, while other developed countries are approximately 70%. Compared with the general level of urbanization in China, the urbanization process in Guizhou Province is relatively fast. Combined with the experience of developed countries, this study sets the annual rate of change to 1% in the benchmark mode, 1.25% in the high-speed mode, and 1% in the high-speed mode. In the low-carbon mode, the annual rate of change was 0.7%.

*3) Setting of per capita GDP*. The per capita GDP of Guizhou Province will continue to grow from 2000 to 2020, with the per capita GDP reaching \$330/person in 2000 and \$7000/person in 2020. In recent years, infrastructure development has led to an increase in economic development in Guizhou Province, and the growth rate of the per capita GDP has rapidly increased. In 2016, the per capita GDP of Guizhou Province increased significantly. According to the 13th Five-Year Plan for National Economic and Social Development of Guizhou Province, the average annual growth rate of the regional GDP has reached 6.6%, and the space for the decline in the per capita GDP growth rate will gradually shrink after the 13th Five-Year Plan. This study set the annual change rate to 6.5% in the benchmark mode, 7% in the high-speed mode, and 6% in the low-carbon mode [36–38, 53, 54].

*4) Residents' consumption levels*. From 2000 to 2020, the average annual growth rate of residents' consumption levels in Guizhou Province was 8.0%. The "13th Five-Year Plan" of Guizhou Province proposes releasing residents' consumption potential, creating consumption demand, and further enhancing their consumption capacity. In baseline mode, the annual change rate was 8%, whereas in high-speed mode, the annual change rate was 9%.

*5) Total energy consumption*. From 2000 to 2020, the total energy consumption in Guizhou Province increased slightly. From 2002 to 2012, total energy consumption showed a rapid upward trend. After 2012, the total energy consumption declined, from 23526 tonnes of standard coal/10,000 CHY in 2012 to 22321 tonnes of standard coal/10,000 CHY in 2018. According to the requirements of energy saving and emission reduction planning in the 13th Five-Year Plan of Guizhou Province, this paper sets the annual change rate of −1.5% in the baseline mode, −2% in the high-speed mode, and −2.5% in the low-carbon mode (Table 13).

## Prediction and analysis of carbon peaks

The fitted WOA-ELM was used to predict the carbon emissions of Guizhou Province from 2020 to 2040 under the three scenarios. The predicted results are listed in Table 14.

**Table 13. Parameter settings for different scenarios.**

| Influencing factors | Profile | Rate of change/ % |
| --- | --- | --- |
| **Total population** | Baseline scenario | 0.70 |
| | High-speed scene | 1.00 |
| | Low-carbon scenarios | 0.50 |
| | Baseline scenario | 1.25 |
| **Urbanization rate** | High-speed scene | 1.50 |
| | Low-carbon scenarios | 1.00 |
| | Baseline scenario | 8.00 |
| **Residents' consumption level** | High-speed scene | 9.00 |
| | Low-carbon scenarios | 7.50 |
| | Baseline scenario | 6.50 |
| **GDP per capita** | High-speed scene | 7.00 |
| | Low-carbon | 6.00 |
| | Baseline scenario | −1.50 |
| **Total energy consumption** | High-speed scene | −2.00 |
| | Low-carbon scenarios | −2.50 |

**Table 14.**

| Profile | Peak carbon emissions (10,000 t) | Peak year for carbon emissions (year) |
|---|---|---|
| **Baseline scenario** | 26243.61 | 2038 |
| **High-speed scene** | 30251.37 | 2036 |
| **Low-carbon scenarios** | 21294.98 | 2033 |

Owing to the differences in the carbon emissions affected by population, urbanization rate, resident consumption level, per capita GDP, and total energy consumption, the occurrence time and peak value of the carbon peak in Guizhou Province will change because of different parameter settings, and the total carbon emissions will also change accordingly. In the baseline scenario, it is estimated that the peak carbon emissions of Guizhou Province will reach 260 million tonnes in 2038, while in the high-speed scenario, they reach 300 million tonnes in 2036 and in the low-carbon scenario they reach 210 million tonnes in 2033. The baseline scenario data shows that the peak carbon emissions in Guizhou Province will not be achieved by 2030 as scheduled and will most likely be delayed to 2038. In the high-speed scenario, the peak carbon emissions in Guizhou Province occurred two years earlier than those in the baseline scenario. By comparing the time and size of the peak carbon emissions under the baseline and low-carbon scenarios, the peak year under the low-carbon scenario was found to be earlier than under the baseline scenario. Although it is three years after the peak target of China in 2030, the development status of Guizhou Province is relatively backward compared to that of developed cities such as Beijing and Shanghai; therefore, it is acceptable. Its peak volume is 40 million tonnes lower than that of the baseline scenario. Comparing the peak time and size of carbon emissions predicted by the low-carbon and high-speed scenarios, shows that the peak time for carbon emissions in the low-carbon scenario is three years earlier than in the high-speed scenario, and the peak volume is reduced by 50 million tonnes. The previous prediction results showed that Guizhou Province cannot achieve the ambitious goal of carbon peak in 2030 in the baseline and high-speed development scenarios. In contrast, the carbon peak time in the low-carbon scenario was earlier and the peak value was lower.

## Discussion

Exploring the main factors affecting carbon emissions in Guizhou Province will be crucial for China to achieve its desired carbon peaks and neutralization. Accurate carbon emission predictions are also of great significance for governments and will help to formulate relevant policies and innovate energy-saving and emission-reduction science and technologies. In this study, the characteristic subset affecting carbon emissions was constructed by referring to the existing literature and combining it with real world data from Guizhou Province. The appropriate input variables were then selected based on the grey correlation analysis method and then the BP neural network and ELM model were established, the WOA algorithm was used to optimize the BP neural network and ELM model, and the performance of the prediction models was compared and analyzed using simulations. Finally, three scenarios were established to predict the carbon emissions of Guizhou Province from 2020 to 2040. The following conclusions were drawn from the analysis.

The results show that the total carbon emissions in Guizhou Province in 2020 will be 22237 million tonnes which was approximately twice the total carbon emissions of Guizhou Province in 2000. With the development of social economy, the growth rate of total carbon emissions in Guizhou Province will gradually decrease, and the overall trend was shown to have an "S"

curve. The data for Guizhou Province and previous studies were combined and 12 influencing factors were selected. According to their degrees of correlation, population and total energy consumption have a greater impact on carbon emissions in Guizhou Province, while the total population, urbanization rate, residents' consumption level, per capita GDP, and total energy consumption were selected as the input variables of the prediction model.

## Conclusion

The BP neural network, ELM, WOA-BP, and WOA-ELM models were established to predict carbon emissions in Guizhou Province. Comparing the mean absolute error, mean absolute percentage error, and root mean square error of the BP neural network, ELM, WOA-BP, and WOA-ELM prediction models, the accuracy of the WOA-ELM model was found to be higher with an MAE of 101. The prediction accuracy of the model based on an extreme learning machine was the second highest, with an MAE of 224. 46, MAPE is 0.43%, RMSE is 328.62, and the prediction effect of the BP neural network model was the worst. Three scenarios were constructed: baseline, high-speed, and low-carbon scenarios. The carbon emissions in Guizhou Province over the next 20 years were determined using the fitted model input with the set scenario parameters. The results show that, under the baseline model, the carbon peak for Guizhou Province will appear in 2038, and the peak value will be 0.61 million tonnes 26243. Under the high-speed scenario, the peak time for carbon emissions in Guizhou Province appeared in 2036, with a peak value is 0.27 million tonnes 30251. Under the low-carbon scenario, the peak time of carbon emissions in Guizhou Province is 2033, and the peak value is 9800 tonnes 21294. In the baseline model, Guizhou Province cannot achieve China's peak target by 2030, and the low-carbon scenario is the closest to the carbon peak target of the three scenarios, indicating that it is necessary to intervene in the external policies of Guizhou Province [55, 56].

According to the results of the above gray correlation analysis, the population has an important impact on carbon emissions in Guizhou Province, which must be considered in the work of energy conservation and emission reduction in Guizhou Province. The increasing demand for energy in daily life and production activities promotes and significantly impacts the increase in carbon emissions. Controlling the population of Guizhou Province and encouraging citizens to choose green travel, energy-saving, and environmentally friendly lifestyles will have far-reaching impacts on the current carbon emissions in Guizhou Province. The government should encourage people to save electricity, appropriately dispose of household appliance waste, and increase investment in the research and development of energy-saving alternatives. Enriching urban public transport, promoting the construction of public transport facilities, and opening more convenient energy vehicle development.

Analysis of the differences in carbon emissions caused by the three development modes in Guizhou Province showed that the low-carbon mode reached the carbon peak earliest, followed by the high-speed, and benchmark modes. In the low-carbon development mode, when carbon dioxide emissions reach their peak, the value is the smallest among the three modes. Overall, population, economic development, and energy consumption factors influence each other, and to reach a timely carbon peak in Guizhou Province, we should not only ensure normal economic growth but also take measures to control the increase in urbanization rate, reduce energy consumption, and optimize energy structures. For example, coal consumption accounts for a high proportion of the energy consumption in Guizhou Province, and coal combustion increases carbon dioxide emissions. Efforts should thus be made to reduce the consumption of coal energy, increase the utilization and conversion rate of coal, increase investment in the research and development of new energy, broaden the scope of the

popularization of new energy, improve the construction of related supporting facilities, and put the full use of new energy on the agenda. There should be a focus on the development of water conservancy, hydropower projects, and photovoltaic projects, and on increasing the proportion of clean energy, such as hydropower.

## Supporting information

**S1 File.**
(DOC)

## Author Contributions

**Conceptualization:** Min Zhang.

**Data curation:** Shi Qiang Yang.

**Formal analysis:** Shi Qiang Yang.

**Methodology:** Wen Rui Ran.

**Project administration:** Wen Rui Ran.

**Supervision:** Min Zhang.

**Validation:** Min Zhang.

**Writing – original draft:** Da Lian.

**Writing – review & editing:** Wu Yang.

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
