## [Decision Letter · Decision Letter 0]

12 Jan 2024

PONE-D-23-42641Prediction of Carbon Emissions in Guizhou Province-Based on Different Neural Network ModelsPLOS ONE

Dear Dr. Yang

Thank you for submitting your manuscript to PLOS ONE. After careful consideration, we feel that it has merit but does not fully meet PLOS ONE’s publication criteria as it currently stands. Therefore, we invite you to submit a revised version of the manuscript that addresses the points raised during the review process. Please submit your revised manuscript by Feb 26 2024 11:59PM. If you will need more time than this to complete your revisions, please reply to this message or contact the journal office at plosone@plos.org. Please include the following items when submitting your revised manuscript:A rebuttal letter that responds to each point raised by the academic editor and reviewer(s). You should upload this letter as a separate file labeled 'Response to Reviewers'.A marked-up copy of your manuscript that highlights changes made to the original version. You should upload this as a separate file labeled 'Revised Manuscript with Track Changes'.An unmarked version of your revised paper without tracked changes. You should upload this as a separate file labeled 'Manuscript'.

We look forward to receiving your revised manuscript.

Kind regards,

Salim Heddam

Academic Editor

PLOS ONE

Journal Requirements:

“This research was supported by Concealed Ore Deposit Exploration and Innovation Team of Guizhou Colleges and Universities (Guizhou Education and Cooperation Talent Team [2015]56), Provincial Key Discipline of Geological Resources and Geological Engineering of Guizhou Province (ZDXK[2018]001), Huang Danian Resources of National colleges and universities Teachers' Team of Exploration Engineering (Teacher Letter [2018] No. 1), Geological Resources and Geological Engineering Talent Base of Guizhou Province (RCJD2018-3), Key Laboratory of Karst Engineering Geology and Hidden Mineral Resources of Guizhou Province (Qianjiaohe KY [2018] No. 486Guizhou Institute of Technology Rural Revitalization Soft Science Project(2022xczx10), Education and Teaching Reform Research Project of Guizhou Institute of Technology (JGZD202107,2022TDFJG01).”

5. We note that your Data Availability Statement is currently as follows: “All relevant data are within the manuscript and its Supporting Information files.”

6. We note that Figure 2 in your submission contain [map/satellite] images which may be copyrighted. All PLOS content is published under the Creative Commons Attribution License (CC BY 4.0), which means that the manuscript, images, and Supporting Information files will be freely available online, and any third party is permitted to access, download, copy, distribute, and use these materials in any way, even commercially, with proper attribution. For these reasons, we cannot publish previously copyrighted maps or satellite images created using proprietary data, such as Google software (Google Maps, Street View, and Earth). For more information, see our copyright guidelines: http://journals.plos.org/plosone/s/licenses-and-copyright.

Additional Editor Comments (if provided):

Reviewer 1#:

It is interesting that prediction of carbon emissions in guizhou province-based on different neural network models is carried out, because global warming caused by greenhouse gas emissions has become a major challenge facingpeople all over the world. The study of regional human activities and their impacts on carbon emis-sions is necessary which can help to achieve the ambitious goal of carbon neutrality and sustainable eco-nomic development.

Suggest reading these literature to expand research ideas, which may include (but are not limited to) the following,https://doi.org/10.1016/j.istruc.2020.12.089. https://doi.org/10.1016/j.jobe.2022.104459.

https://doi.org/10.1007/s12205-023-0677-9.https://doi.org/10.1016/j.istruc.2023.02.048.

https://doi.org/10.1007/s11069-019-03709-x. https://doi.org/10.1016/j.engstruct.2019.109500.

https://doi.org/10.1016/j.engstruct.2020.110434.https://doi.org/10.2749/101686616X1311.

It is noted that your manuscript needs careful editing by someone with expertise in technical English editing paying particular attention to English grammar, spelling, and sentence structure so that the goals and results of the study are clear to the reader.

Reviewer 2#:

Global warming caused by greenhouse gas emissions has become a major challenge facing people all over the world. The study of regional human activities and their impacts on carbon emissions is of great significance to achieve the ambitious goal of carbon neutrality and sustainable economic development. Guizhou Province is a typical karst area in China, and its energy consumption is mainly based on fossil fuels. Therefore, it is necessary to predict and analyze its carbon emissions. In this paper, BP neural network and extreme learning machine (ELM) model, which have the advantage of nonlinear processing, will be used to predict the carbon emissions of Guizhou Province from 2020 to 2040. Based on the energy consumption data of Guizhou Province, the carbon emissions of Guizhou Province are calculated by using the conversion method and the inventory compilation method. Overall, I think this manuscript is suitable for this journal’s scope.

However, there are some issues that may need to be improved.

1 Abstract. The review part should be more concise, and the findings part should be more accurate and detailedly. The summary can be accurately summarized in several points

2 Tons should be tonnes

3 Introduction: The novelty of this paper should be further justified by highlighting main contributions to the existing introduction and literature review. For example, what are the other researches on the prediction model? There are many articles, e.g., one about the ARIMA method entitled “Ecological footprint simulation and prediction by ARIMA model-A case study in Henan Province of China. https://doi.org/ 10.1016/j.ecolind.2009.06.007”.

4 Yuan should be CHY

5 The formula part of the method of the full text, as detailed and specific as possible, at present, the formula is few and unclear

6 There are still unclear in the graphics

7 Grammar problems. Need to check the polish carefully

8 There should be a conclusion. The conclusion can also be clearly and concisely divided into several points.

9 The conclusion can be preceded by a discussion section. The section should enhance the relevant discussion. For example, the comparison of the results of this paper with those of its predecessor could be strengthened with regard to the results of carbon emissions in China or World or GUIZHOU. One China’s article entitled “China's CO2 emissions: A systematical decomposition concurrently from multi-sectors and multi-stages since 1980 by an extended logarithmic mean divisia index. https://doi.org/10.1016/j.esr.2023.101141”. Next world one entitled “Contribution of Renewable Energy Consumption to CO2 Emission Mitigation: A Comparative Analysis from a Global Geographic Perspective, https://doi.org/ 10.3390/su13073853”.

Reviewer 3#:

This manuscript proposes an improved neural network model to predict the future carbon emissions in Guizhou Province. Currently, limited research uses machine learning methods to predict carbon emissions, and most studies use relatively simple neural networks. Directly training neural networks with backpropagation algorithms can lead to local optima, but this manuscript combines neural networks with various optimization strategies to enhance generalization capabilities. The training dataset consists of carbon emissions in Guizhou Province from 2000 to 2020, calculated using the inventory method. Twelve relevant descriptors were initially selected, with the five most correlated ones chosen as inputs to improve prediction accuracy. Several improved neural network models were designed, including WOA-BP and WOA-ELM models, to provide a more reasonable initial weight for the neural network to avoid overfitting. Compared to the naive BP neural network, the improved WOA-ELM model significantly enhances carbon emission prediction, validating its effectiveness. To comprehensively analyze carbon emissions in Guizhou Province, three carbon emission scenarios were constructed, and using the best-performing WOA-ELM model, the paper predicted the years of reaching carbon emission peaks under different scenarios. Based on the prediction results, the authors also provided reasonable suggestions for reducing Guizhou's carbon emissions.

The results of this work are substantial and innovative in the field, and the paper is recommended for acceptance after addressing the following points:

(1) There are instances of colloquial English expressions and language issues. Please review carefully. For example, the phrase "collects the relevant data" in line 372 and "Setting of model input layer and output layer" in line 397, and more.

(2) The motivation for using ELM and WOA to enhance the neural network must be established. It would be beneficial to elaborate on why these optimization strategies were chosen and whether they have been used in the field.

(3) It is necessary to provide the training curves and convergence of the machine learning models to ensure they do not suffer overfitting on such a limited dataset.

Reviewers' comments:

Reviewer's Responses to Questions

**Comments to the Author**

1. Is the manuscript technically sound, and do the data support the conclusions?

Reviewer #1: No

Reviewer #2: Partly

Reviewer #3: Yes

2. Has the statistical analysis been performed appropriately and rigorously? 

Reviewer #1: No

Reviewer #2: I Don't Know

Reviewer #3: Yes

3. Have the authors made all data underlying the findings in their manuscript fully available?

Reviewer #1: Yes

Reviewer #2: Yes

Reviewer #3: Yes

4. Is the manuscript presented in an intelligible fashion and written in standard English?

Reviewer #1: No

Reviewer #2: Yes

Reviewer #3: Yes

5. Review Comments to the Author

Reviewer #1: It is interesting that prediction of carbon emissions in guizhou province-based on different neural network models is carried out, because global warming caused by greenhouse gas emissions has become a major challenge facingpeople all over the world. The study of regional human activities and their impacts on carbon emis-sions is necessary which can help to achieve the ambitious goal of carbon neutrality and sustainable eco-nomic development.

Suggest reading these literature to expand research ideas, which may include (but are not limited to) the following,https://doi.org/10.1016/j.istruc.2020.12.089. https://doi.org/10.1016/j.jobe.2022.104459.

https://doi.org/10.1007/s12205-023-0677-9.https://doi.org/10.1016/j.istruc.2023.02.048.

https://doi.org/10.1007/s11069-019-03709-x. https://doi.org/10.1016/j.engstruct.2019.109500.

https://doi.org/10.1016/j.engstruct.2020.110434.https://doi.org/10.2749/101686616X1311.

It is noted that your manuscript needs careful editing by someone with expertise in technical English editing paying particular attention to English grammar, spelling, and sentence structure so that the goals and results of the study are clear to the reader.

Reviewer #2: Global warming caused by greenhouse gas emissions has become a major challenge facing people all over the world. The study of regional human activities and their impacts on carbon emissions is of great significance to achieve the ambitious goal of carbon neutrality and sustainable economic development. Guizhou Province is a typical karst area in China, and its energy consumption is mainly based on fossil fuels. Therefore, it is necessary to predict and analyze its carbon emissions. In this paper, BP neural network and extreme learning machine (ELM) model, which have the advantage of nonlinear processing, will be used to predict the carbon emissions of Guizhou Province from 2020 to 2040. Based on the energy consumption data of Guizhou Province, the carbon emissions of Guizhou Province are calculated by using the conversion method and the inventory compilation method. Overall, I think this manuscript is suitable for this journal’s scope.

However, there are some issues that may need to be improved.

1 Abstract. The review part should be more concise, and the findings part should be more accurate and detailedly. The summary can be accurately summarized in several points

2 Tons should be tonnes

3 Introduction: The novelty of this paper should be further justified by highlighting main contributions to the existing introduction and literature review. For example, what are the other researches on the prediction model? There are many articles, e.g., one about the ARIMA method entitled “Ecological footprint simulation and prediction by ARIMA model-A case study in Henan Province of China. https://doi.org/ 10.1016/j.ecolind.2009.06.007”.

4 Yuan should be CHY

5 The formula part of the method of the full text, as detailed and specific as possible, at present, the formula is few and unclear

6 There are still unclear in the graphics

7 Grammar problems. Need to check the polish carefully

8 There should be a conclusion. The conclusion can also be clearly and concisely divided into several points.

9 The conclusion can be preceded by a discussion section. The section should enhance the relevant discussion. For example, the comparison of the results of this paper with those of its predecessor could be strengthened with regard to the results of carbon emissions in China or World or GUIZHOU. One China’s article entitled “China's CO2 emissions: A systematical decomposition concurrently from multi-sectors and multi-stages since 1980 by an extended logarithmic mean divisia index. https://doi.org/10.1016/j.esr.2023.101141”. Next world one entitled “Contribution of Renewable Energy Consumption to CO2 Emission Mitigation: A Comparative Analysis from a Global Geographic Perspective, https://doi.org/ 10.3390/su13073853”.

Reviewer #3: This manuscript proposes an improved neural network model to predict the future carbon emissions in Guizhou Province. Currently, limited research uses machine learning methods to predict carbon emissions, and most studies use relatively simple neural networks. Directly training neural networks with backpropagation algorithms can lead to local optima, but this manuscript combines neural networks with various optimization strategies to enhance generalization capabilities. The training dataset consists of carbon emissions in Guizhou Province from 2000 to 2020, calculated using the inventory method. Twelve relevant descriptors were initially selected, with the five most correlated ones chosen as inputs to improve prediction accuracy. Several improved neural network models were designed, including WOA-BP and WOA-ELM models, to provide a more reasonable initial weight for the neural network to avoid overfitting. Compared to the naive BP neural network, the improved WOA-ELM model significantly enhances carbon emission prediction, validating its effectiveness. To comprehensively analyze carbon emissions in Guizhou Province, three carbon emission scenarios were constructed, and using the best-performing WOA-ELM model, the paper predicted the years of reaching carbon emission peaks under different scenarios. Based on the prediction results, the authors also provided reasonable suggestions for reducing Guizhou's carbon emissions.

The results of this work are substantial and innovative in the field, and the paper is recommended for acceptance after addressing the following points:

(1) There are instances of colloquial English expressions and language issues. Please review carefully. For example, the phrase "collects the relevant data" in line 372 and "Setting of model input layer and output layer" in line 397, and more.

(2) The motivation for using ELM and WOA to enhance the neural network must be established. It would be beneficial to elaborate on why these optimization strategies were chosen and whether they have been used in the field.

(3) It is necessary to provide the training curves and convergence of the machine learning models to ensure they do not suffer overfitting on such a limited dataset.

6. PLOS authors have the option to publish the peer review history of their article (what does this mean?). If published, this will include your full peer review and any attached files.

Reviewer #1: No

Reviewer #2: No

Reviewer #3: **Yes: **Jia-Ji Zhu

---

## [Author Response · Author response to Decision Letter 0]

19 Apr 2024

Dear editor：

Thank you very much for your letter. We have learned much from your and three reviewers’ comments, which are fair, encouraging and constructive. After carefully studying the comments and your advice, we have made corresponding changes. The main revisions are listed below.

For reviewer one:

1.It is noted that your manuscript needs careful editing by someone with expertise in technical English editing paying particular attention to English grammar, spelling, and sentence structure so that the goals and results of the study are clear to the reader

. We have carefully revised the article and edited it in English in the attachment

2.Suggest reading these literature to expand research ideas

Thank you very much. This is a great recommendation. We have studied these high-quality articles carefully, and have been inspired by their ideas. We have also made targeted modifications to the articles and added them to the references

For reviewer two:

1 Abstract. The review part should be more concise, and the findings part should be more accurate and detailedly. The summary can be accurately summarized in several points

The abstract has been streamlined and modified

2 Tons should be tonnes

Conducted a full text review and made modifications

3 Introduction: The novelty of this paper should be further justified by highlighting main contributions to the existing introduction and literature review. For example, what are the other researches on the prediction model? There are many articles, e.g., one about the ARIMA method entitled “Ecological footprint simulation and prediction by ARIMA model-A case study in Henan Province of China. https://doi.org/ 10.1016/j.ecolind.2009.06.007”.

This is a great recommendation. We have studied the article carefully, made modifications and adjustments to this section, and also cited the literature

4 Yuan should be CHY

Conducted a full text review and made modifications

5 The formula part of the method of the full text, as detailed and specific as possible, at present, the formula is few and unclear

Made modifications and adjustments

6 There are still unclear in the graphics

Unclear images removed

7 Grammar problems. Need to check the polish carefully

Made modifications and adjustments

8 There should be a conclusion. The conclusion can also be clearly and concisely divided into several points.

9 The conclusion can be preceded by a discussion section. The section should enhance the relevant discussion. For example, the comparison of the results of this paper with those of its predecessor could be strengthened with regard to the results of carbon emissions in China or World or GUIZHOU. One China’s article entitled “China's CO2 emissions: A systematical decomposition concurrently from multi-sectors and multi-stages since 1980 by an extended logarithmic mean divisia index. https://doi.org/10.1016/j.esr.2023.101141”. Next world one entitled “Contribution of Renewable Energy Consumption to CO2 Emission Mitigation: A Comparative Analysis from a Global Geographic Perspective, https://doi.org/ 10.3390/su13073853”

We have added these two parts and made additions and modifications. Carefully read the provided references. This is constructive feedback, thank you for the reviewer's suggestions.we also cited the article

For reviewer three:

1  There are instances of colloquial English expressions and language issues. Please review carefully

Made modifications and adjustments

2 The motivation for using ELM and WOA to enhance the neural network must be established. It would be beneficial to elaborate on why these optimization strategies were chosen and whether they have been used in the field.

An explanation was added to the text:It is worth noting that in ELM, the input data is transformed by the hidden layer, and then the output layer produces the result. This process is "forward propagation", that is, information flows from the input layer to the output layer. However, the most important thing in this process is backpropagation, that is, how to adjust network parameters to improve performance when the output does not meet expectations. Back-propagation algorithm is an important optimization technique, which calculates how much the weight of each layer needs to be adjusted according to the difference between the actual output and the expected output of the network, and then optimizes the network. In ELM, due to its single-layer feedforward feature, backpropagation is mainly used to adjust the weights and biases, so that the network can better adapt to the training data. Whale optimization algorithm WOA is used to optimize the learning rate of convolutional neural network CNN and the number of hidden layer neural networks. It has higher optimization efficiency and better optimization results. Especially when the data used is only a single column of time series data, WOA algorithm may show better optimization results when dealing with more complex data. Therefore, the WOA algorithm is expected to become an important tool in the optimization of convolutional neural networks.

3  It is necessary to provide the training curves and convergence of the machine learning models to ensure they do not suffer overfitting on such a limited dataset.

Convergence explanation has been added to the article

Eg:

1.When its value is small, although it can make the simulation results of the model more accurate, it significantly increases the training time. In general studies, the learning rate γ is usually set between 0 and 1. In this paper, through continuous debugging and comparison of the training effect in the training process, the learning rate γ = 0.1 was selected. The accuracy of the network training was required to be 0.001, and the maximum number of training sessions was 500.

2.When setting the initial weights and thresholds of the neural network, a set of randomly generated initial values was selected, because there was no relevant setting principle. The BP neural network can learn the mapping relationship between the input and output automatically, generate initial parameters randomly, and modify the weights and thresholds of the network continuously through error back propagation; however, randomly selected initial weights and thresholds are usually inversely proportional to the convergence speed of neural network training; that is, the larger the value, the slower the convergence speed. In this case, the final training results easily fall into the local optimum, and it is difficult to obtain ideal calculation and prediction results.

At the same time, the data table is given in the paper.

Thank you to the judges for their hard work. During the traditional Chinese Spring Festival, I also sincerely wish everyone a happy holiday

Yang Wu

---

## [Decision Letter · Decision Letter 1]

14 May 2024

Carbon peaking prediction scenarios based on different neural network models: A case study of Guizhou Province

PONE-D-23-42641R1

Dear Dr. Yang

We’re pleased to inform you that your manuscript has been judged scientifically suitable for publication and will be formally accepted for publication once it meets all outstanding technical requirements.

Kind regards,

Salim Heddam

Academic Editor

PLOS ONE

Additional Editor Comments (optional):

Reviewer 1#:

After revisions, the introduction in this manuscript provide sufficient background and include all relevant references,and the research design is appropriate.What's more, the methods are adequately described and the results are clearly presented, so the conclusions are supported by the results.The analyses and the results reported in this manuscript are interesting and useful for future researches and preliminary design.

In my opinion,it can be accepted in present form.

Reviewer 2#: The author should revise and polish the paper according to the format of the journal.

Reviewer 3#:

Reviewers' comments:

Reviewer's Responses to Questions

**Comments to the Author**

1. If the authors have adequately addressed your comments raised in a previous round of review and you feel that this manuscript is now acceptable for publication, you may indicate that here to bypass the “Comments to the Author” section, enter your conflict of interest statement in the “Confidential to Editor” section, and submit your "Accept" recommendation.

Reviewer #1: All comments have been addressed

Reviewer #2: All comments have been addressed

Reviewer #3: All comments have been addressed

2. Is the manuscript technically sound, and do the data support the conclusions?

Reviewer #1: Yes

Reviewer #2: Yes

Reviewer #3: Yes

3. Has the statistical analysis been performed appropriately and rigorously? 

Reviewer #1: Yes

Reviewer #2: Yes

Reviewer #3: Yes

4. Have the authors made all data underlying the findings in their manuscript fully available?

Reviewer #1: Yes

Reviewer #2: Yes

Reviewer #3: Yes

5. Is the manuscript presented in an intelligible fashion and written in standard English?

Reviewer #1: Yes

Reviewer #2: Yes

Reviewer #3: Yes

6. Review Comments to the Author

Reviewer #1: After revisions, the introduction in this manuscript provide sufficient background and include all relevant references,and the research design is appropriate.What's more, the methods are adequately described and the results are clearly presented, so the conclusions are supported by the results.The analyses and the results reported in this manuscript are interesting and useful for future researches and preliminary design.

In my opinion,it can be accepted in present form.

Reviewer #2: The author should revise and polish the paper according to the format of the journal.

The author should revise and polish the paper according to the format of the journal.

The author should revise and polish the paper according to the format of the journal.

Reviewer #3: (No Response)

7. PLOS authors have the option to publish the peer review history of their article (what does this mean?). If published, this will include your full peer review and any attached files.

Reviewer #1: No

Reviewer #2: No

Reviewer #3: No

---

## [Editor Report · Acceptance letter]

22 May 2024

PONE-D-23-42641R1 

PLOS ONE

Dear Dr. Yang, 

I'm pleased to inform you that your manuscript has been deemed suitable for publication in PLOS ONE. Congratulations! Your manuscript is now being handed over to our production team.

Kind regards, 

on behalf of

Dr. Salim Heddam 

Academic Editor

PLOS ONE